# β-Sitosterol Attenuates High Grain Diet-Induced Inflammatory Stress and Modifies Rumen Fermentation and Microbiota in Sheep

**DOI:** 10.3390/ani10010171

**Published:** 2020-01-19

**Authors:** Guangliang Xia, Jie Sun, Yaotian Fan, Fangfang Zhao, Gulzar Ahmed, Yaqian Jin, Ying Zhang, Hongrong Wang

**Affiliations:** Laboratory of Metabolic Manipulation of Herbivorous Animal Nutrition, College of Animal Science and Technology, Yangzhou University, Yangzhou 225009, Chinad160085@yzu.edu.cn (F.Z.);

**Keywords:** β-sitosterol, subacute ruminal acidosis, inflammatory response, bacterial community, pyrosequencing, sheep

## Abstract

**Simple Summary:**

Over the years, rumen acidosis is considered as the most common disease in ruminants, which affects animal health and brings huge economic losses of the ruminant industry. Antibiotic have been considered as effective to alleviate the occurrence of ruminal acidosis. However, antibiotic usage in food animals has been absolutely banned by European Union and strictly restricted in other countries. It is necessary to search for safe, effective and healthy additives alternative to antibiotics for ruminants. β-sitosterol is a plant compound known as phytosterol, which has many biological activities. In this study, sheep were feed four levels with β-sitosterol supplementation (0, 0.25, 0.5, and 1.0 g/kg). We concluded that β-sitosterol could attenuate high grain diet-induced inflammatory response and modify ruminal fermentation. These findings provide updated insight for preventing the occurrence of rumen acidosis.

**Abstract:**

β-sitosterol (BSS) is a plant-derived natural bioactive compound, its cellular mechanism of anti-inflammatory activity has been proven recently. Little information is available regarding the application of BSS on ruminants under high grain diet. The objective of this study was to evaluate the effects of dietary BSS supplementation on inflammatory response, ruminal fermentation characteristics and the composition of the ruminal bacterial community under high grain diet. Eight rumen-cannulated Hu sheep (59.7 ± 4.8 kg of initial body weight) were randomly assigned into a replicated 4 × 4 Latin square design trial. Sheep were fed a high grain diet (non-fiber carbohydrate: neutral detergent fiber = 2.03) supplemented either with 0.25 (LBS), 0.5 (MBS), 1.0 (HBS) or without (CON) g BSS /kg dry matter diet. On day 21 of each period, rumen content samples were obtained at 6 h postfeeding, and blood samples were obtained before morning feeding. The data showed that compared with control group, Dietary BSS supplementation decreased serum concentrations of tumor necrosis factor, interleukin (IL)-6, and IL-1β. The ruminal pH and acetate concentration for BSS treatment were improved, while concentration of propionate, butyrate and lactate was decreased. The result of Illumina MiSeq sequencing of 16S rRNA gene revealed that BSS addition can increase the proportion of *Prevotella_1*, *Rikenellaceae_RC9_gut_group*, *Prevotella_7*, and *Selenomonas_1*, and decrease the proportion of *Lachnospiraceae_NK3A20_group*. These results indicated that BSS attenuates high grain diet-induced inflammatory response and modifies ruminal fermentation. In addition, the BSS dietary supplementation at the level of 0.5 g/kg is recommended in sheep.

## 1. Introduction

Rumen acidosis is the consequence of feeding highly digestible grain diets and it is considered the most common nutritional disorder in feedlot cattle [1]. Long-term feeding of high grain diets is associated with subacute ruminal acidosis (SARA), induces changes in bacterial community composition and metabolic disorders. Ruminal pH is reduced lower than 5.8 during SARA due to the decreased in the abundances of cellulolytic bacteria and the increased in the proportions of starch-fermenting and lactic acid producing bacteria [2]. The lower pH in the rumen content may enhance the lysis of gram-negative bacteria which is accompanied with increasing ruminal free lipopolysaccharide (LPS) [3]. However, following the rumen epithelial barrier failed in a low-pH rumen condition, LPS may be translocated into the interior circulation of the body and triggered systemic inflammation by promoting the release of pro-inflammatory cytokines, such as tumor necrosis factor (TNF)-α, interleukin (IL)-1β and IL-6 [4,5].

Several diet strategies, such as sodium bicarbonate buffer [6], dicarboxylic acids [7], monensin iono-phones [8], and probiotics [9], have been used to prevent the occurrence of SARA. However, antibiotic usage in food animals has been absolutely banned by the European Union and strictly restricted in other countries due to public concerns about food safety and antibiotic resistance [10]. Currently, dietary supplementation with plant-derived extracts, for example, alkaloid [11], terpenoid [12], and essential oils [13], has received widespread attention from researchers due to their ability to stabilize rumen pH and improve rumen fermentation in ruminants.

Phytosterols are plant-derived natural bioactive compounds, mainly including β-sitosterol (BSS), stigmasterol, campesterol, and brassicasterol [14]. β-sitosterol is the most abundant phytosterol, and it has been shown to reduce the absorption of cholesterol and the serum level of cholesterol in previous studies. It has been reported that BSS has many biological activities, such as cholesterol-lowering [15], anticancer [16], antioxidant [17], and anti-inflammatory [18]. Furthermore, several studies have been demonstrated that BSS ameliorated high-fat diet-induced intestinal inflammation in mice [19], suppressed the secretion of inflammatory factors from keratinocytes and macrophages induced by LPS, such as TNF-α, IL-1β, IL-6, and IL-8, respectively [20]. In spite of the excellent aforementioned biological functions, information is extremely scarce regarding dietary BBS supplementation on inflammatory response and ruminal bacterial community in ruminants fed high grain-diets. The aim of the present experiment was to investigate the effects of dietary BSS supplementation at different levels on the ruminal fermentation, inflammatory factors and ruminal microbial community under high grain diet.

## 2. Materials and Methods

### 2.1. Animals Diets and Experimental Design

Animal care and procedures were in accordance with the Chinese guidelines for animal welfare and carried out with permission of the Animal Care and Use Committee of Yangzhou University (SXXY2015-0054). Eight Hu sheep (59.7 ± 4.8 kg of initial body weight) fitted with ruminal cannula were allocated to a replicated 4 × 4 Latin square design (n = 8). Four dietary treatments were used: A high-grain diet without BSS supplementation (CON; Table 1) and supplemented with 0.25 (LBS), 0.5 (MBS), or 1 (HBS) g BSS/kg dry matter intake (DMI). The experimental diets were formulated according to Chinese feeding standards for meat producing sheep of meat producing sheep (NY/T 816-2004) [21] to meet or exceed the energy requirements of Hu sheep rams. BSS (purity ≥ 95%; Cool Chemical Technology Co., Ltd., Beijing, China) was administered via the rumen cannula twice daily after feeding. The diet (1.8 kg dry matter (DM) per sheep per day) was offered in equal amounts at 08:00 and 18:00 daily and 5% to 10% of feed refusals were allowed. The sheep were placed in individual pens (2 × 1.5 m) with free access to water. Four periods were included and each experimental period consisted of 21 days. To allow better adaptation to the high grain diet, the proportion of grain was incrementally increased during the first 7 days of each period. Samples of diets and refusals were collected daily and analyzed as described by Pan et al. [22]. 

### 2.2. Rumen Fluid Sampling and Analysis

On day 21 of each period, rumen contents were sampled from different positions in the cranial and ventral sacs of the rumen, pH was determined every 3 h from 0 to 12 h after the morning. Rumen contents were sampled at 6 h postfeeding were strained through four layers of cheesecloth into a glass beaker. The strained rumen fluid samples were centrifuged immediately at 10,000× *g* for 15 min at 4 °C. The supernatant was passed through a disposable 0.22 μm sterile, pyrogen-free filter. Subsamples were taken and further heated at 100 °C for 30 min before being stored at −20 °C for LPS determination. LPS concentration was determined using a Chromogenic End-point Tachypleus Amebocyte Lysate Assay Kit (Chinese Horseshoe Crab Reagent Manufactory Co., Ltd., Xiamen, China).

Samples (20 mL) of strained rumen fluid were centrifuged at 2500× *g* for 15 min at 4 °C and the supernatants were stored at −20 °C for analysis of volatile fatty acid (VFA), lactate and ammonia nitrogen (NH_3_-N). Another portion of 20 mL rumen fluid was stored at −80 °C immediately for microbial DNA extraction.

The VFA concentrations were determined by gas chromatography (GC-14B, shimadzu, Kyoto, Japan; film thickness of the capillary column, 30 m × 0.53 m × 1 µm; column temperature, 110 °C; injector and detector temperature, 200 °C) with some modifications. In brief, the supernatants of rumen fluid were thawed at 4 °C and 0.2 mL of 20% metaphosphoric acid solution containing 60 mM crotonic acid was added to 1 mL supernatant. The tubes were vortexed and centrifuged at 10,000× *g* for 10 min at 4 °C and 1 mL of supernatants were into gas chromatography for analysis. The concentration of lactate was measured as described by Barker et al. [23]. NH_3_-N concentration of rumen fluid was analyzed as described by Bhandari et al. [24].

### 2.3. Blood Sampling and Analysis

Before morning feeding on day 21 of each period, jugular blood was sampled in blank and heparinized 10-mL evacuated tubes for serum and plasma collection, respectively. Samples were immediately kept on ice until being centrifuged at 3500× *g* for 15 min at 4 °C. The serum was divided into 2 mL aliquots and stored at −20 °C for the analysis of TNF-α, IL-6, IL-8, IL-1β, and Toll-like receptor 4 (TLR4). A portion of plasma was transferred to pyrogen-free glass tubes and stored at −20 °C for LPS analysis.

The serum concentrations of TNF-α, IL-6, IL-8, IL-1β, and TLR4 were determined using ELISA kits (TNF-α: YFXEG00001; IL-6: YFXEG00002; IL-1β: YFXEG00003; IL-1β, YFXEG00005; TLR4, YFXEG00006; YIFEIXUE BIO TECH, Nanjing, Jiangsu, China) according to the manufacturer’s instructions, respectively. The concentrations of LPS in plasma were determined by a chromogenic end-point LAL assay (Chinese Horseshoe Crab Reagent Manufactory Co., Ltd., Xiamen, China) with a minimum detection limit of 0.1 EU/mL.

### 2.4. DNA Extraction, Illumina MiSeq Sequencing and Sequencing Data Processing

Rumen samples taken at 6 h after feeding were thawed at room temperature and a 1 mL aliquot was centrifuged at 10,000× *g* for min at 4 °C and the supernatant was discarded. DNA was extracted from the pellets using a QIAamp DNA Stool Mini Kit (Qiagen, Hilden, Germany) which include a bead-beating step for the mechanical lysis of the microbial cells. DNA concentration and purity were determined spectrophotometrically by measuring the A_260/280_ (Beckman DU/800; Beckman Coulter, Inc., Fullerton, CA, USA). DNA was stored at −80 °C until further processing.

Next generation sequencing library preparations and Illumina MiSeq sequencing were conducted at GENEWIZ Inc. (Suzhou, China). Qubit 2.0 Fluorometer (Invitrogen, Carlsbad, CA, USA) was used for quantifying DNA samples and 0.8% agarose gel was used for check the DNA quality. Between 30 and 50 ng DNA was used to generate amplicons using a MetaVx™ Library Preparation kit (GENEWIZ Inc., South Plainfield, NJ, USA). The forward primers containing the sequence ‘‘CCTACGGRRBGCASCAGKVRVGAAT” and reverse primers containing the sequence ‘‘GGACTACNVGGGTWTC TAATCC” was used for amplifying the V3 and V4 regions. Besides the 16S target-specific sequence, the primers also contain adaptor sequences allowing uniform amplification of the library with high complexity ready for downstream NGS sequencing on Illumina MiSeq.

Agilent 2100 Bioanalyzer (Agilent Technologies, Palo Alto, CA, USA) was used for validating the DNA libraries, Qubit and real time PCR (Applied Biosystems, Carlsbad, CA, USA) were used for quantifying the DNA libraries. DNA libraries were multiplexed and loaded on an Illumina MiSeq instrument according to manufacturer’s instructions (Illumina, San Diego, CA, USA). Sequencing was performed using a 2 × 250 or 2 × 300 paired-end (PE) configuration, image analysis and base calling were conducted by the MiSeq Control Software (MCS) on the MiSeq instrument. The sequences were processed and analyzed by GENEWIZ. Taxonomy analysis was carried out on Qiime platform.

Sequence data were analyzed on a Qiime platform using default parameters [25]. Quality filters were applied to select sequences for OTU analysis: more than 200 bp in length, absence of vague base ‘N’ and average base quality score higher than 25. High-quality sequences were clustered into operational taxonomic units (OTUs) using VSEARCH (1.9.6) [26] with 97% sequence identity threshold. All the OTUs were assigned taxonomic categories using the ribosomal database program (RDP) classifier at a confidence threshold of 0.8 and then predicted to the species level using the Silva132 database. Community diversity was estimated with the normalized reads using the ACE (abundance-based coverage estimator), Chao1, and Shannon indices. 

### 2.5. Statistical Analysis

Blood parameters, rumen fermentation parameters, diversity index, and bacterial abundance were subjected to an ANOVA analysis, using GLM of SAS (version 9.2, SAS Institute Inc., Cary, NC, USA). The experimental data were analyzed in a replicated 4 × 4 Latin square design, using following model:Y_ijkl_ = μ + T_i_ + P_j_ + C_k_ + ε_ijk_
where Y_ijklm_ is the response variable value of the kth sheep subjected to the ith treatment in the jth period, μ is the overall mean, T_i_ is fixed effect of the treatments (i = 1, 2, 3, and 4), P_j_ is the fixed effect of period (j = 1, 2, 3, and 4), C_k_ is the random effect of the kth sheep, and ε_ijk_ is the random residual error. Period × treatment interactions were initially included and found to be very small, and they were finally removed from the model. Pearson correlations between bacterial communities and ruminal fermentation variables were analyzed using the PROC CORR procedure of SAS. Only those bacterial taxa with an abundance ≥ 0.1% of the total community in at least one ruminal sample were included in the analysis. Significance was declared at *p* < 0.05.

## 3. Results

### 3.1. Induction of SARA Model

As shown in Figure 1, higher pH values were observed in sheep that received treatment HBS than in the control at 3 h and 6 h after morning feeding (*p* < 0.05). Compared with the control group, higher ruminal pH values were observed in MBS group at 3 h and 6 h after morning feeding, but the difference was not significant (*p* > 0.05). There was no significant difference in ruminal pH between LBS group and CON group from 0 to 12 h after morning feeding. In addition, ruminal pH in sheep fed the CON diet was below 5.8 for more than 3 h/day (from 3 h to 12 h sampling times).

### 3.2. Blood Parameters

As shown in Table 2, compared with the CON group, the concentrations of LPS, TNF-α, IL-6, and IL-1β were decreased (*p* < 0.05) in MBS and HBS treatments, whereas the concentration of TLR4 increased (*p* < 0.05) in LBS treatment. There were no apparent variations (*p* > 0.05) in the concentration of IL-8 between the control and BSS treatments. 

### 3.3. Ruminal Fermentation Characteristics 

As shown in Table 3, compared with the CON treatment, the HBS treatment increased (*p* < 0.05) ruminal pH value and decreased the molar proportions of propionate. The MBS and HBS treatments increased (*p* < 0.05) the concentrations of NH_3_-N, and the molar proportions of acetate, whereas the concentrations of lactate and LPS were decreased significantly compared with control group (*p* < 0.05). The MBS treatment presented with marked decreases (*p* < 0.05) in the molar proportions of butyrate and valerate compared that with the control. No significant differences (*p* > 0.05) was observed in the molar proportions of isobutyrate, isovalerate and TVFA between the control and BSS treatments. 

### 3.4. Diversity and Structure of the Ruminal Microbiota Composition

A total of 2,105,304 raw reads were detected from bacterial sequencing. Quality filtering and purifying chimeric resulted in 1,805,287 high quality sequences, which clustered in 722 OTUs with 56,415 reads per sample. A Venn diagram revealed that the numbers of unique OUT for the CON, LBS, MBS, and HBS groups were 15, 18, 23, and 17, respectively (Figure 2a). Figure 2b demonstrated that rarefaction curve became gentle after 5000 reads, which indicated that almost all bacterial species detected were observed in each sample. Overall, 12 phyla were identified in ruminal samples. According to the average sequence number, Bacteroidetes (61.63%), Firmicutes (30.48%), and Proteobacteria (5.89%) were the three dominant phyla, which composed more than 98% of the ruminal microbiota (Figure 3).

In terms of alpha bacterial diversity (Table 4), the LBS treatment decreased the OTU numbers compared with the CON treatment. No differences were observed in the ACE, Chao 1, Shannon, Simpson index and Good’s coverage between the control and BSS treatments.

### 3.5. Effect of BSS Supplementation on Relative Abundance of Bacterial Communities

At the phylum level (Table 5), compared with the control, the MBS treatment increased (*p* < 0.05) the relative abundance of Bacteroidetes and decreased (*p* < 0.05) the proportions of Cyanobacteria, whereas the HBS treatment only decreased the relative abundance of Cyanobacteria. There were no significant shifts (*p* > 0.05) detected in the relative abundance of Firmicutes, Actinobacteria, Patescibacteria, Euryarchaeota, Proteobacteria, Tenericutes, Spirochaetes, k_Bacteria_Unclassified, Chloroflexi, and Synergistetes.

Of the 53 genus observed with over 0.1% of relative abundance, 20 changed significantly between the control and BSS treatments (Table 6). Compared with the control, the abundance of *f__Prevotellaceae_Unclassified, Erysipelotrichaceae_UCG-009*, *Shuttleworthia, Syntrophococcus,* and *Lactobacillus* were significantly increased (*p* < 0.05) in the LBS group and the relative abundance of *Prevotella_1, Prevotella_7*, *Prevotellaceae_UCG004, Erysipelotrichaceae_UCG-004, Succiniclasticum,* and *Schwartzia* were significantly increased in the MBS group. The HBS treatment presented with marked increases in the abundance levels of *Prevotella_1*, *Rikenellaceae_RC9_gut_group*, *Selenomonas_1*, *Prevotellaceae_UCG-003*, and *Ruminococcaceae_UCG002* (*p* < 0.05) compared that with the control. The MBS and HBS groups decreased (*p* < 0.05) the relative abundance of Erysipelotrichaceae_UCG006, *f__Erysipelotrichaceae_Unclassified,* and *Howardella*, and the HBS group decreased (*p* < 0.05) the relative abundance of *Acetitomaculum* compared with the control. 

### 3.6. Correlations between Bacterial Communities and Ruminal Variables

As shown in Figure 4, a total of nine genera were negatively correlated with rumen NH_3_-N concentrations (*p* < 0.05), namely *Syntrophococcus*, *Prevotella_7*, *Prevotellaceae_UCG004*, *Schwartzia*, *f__Prevotellaceae_Unclassified*, *Erysipelotrichaceae_UCG009*, Shuttleworthia, *Lactobacillus*, and *Erysipelotrichaceae_UCG006*. The molar proportions of acetate and TVFA were positively correlated to the relative abundance of *Prevotella_1*, *Prevotella_7*, *Prevotellaceae_UCG004,* and *Shuttleworthia* (*p* < 0.05), but negatively correlated to *f__Erysipelotrichaceae_Unclassified* (*p* < 0.05). In addition, the genus *Acetitomaculum* and *Lachnospiraceae_NK3A20_group* were positively correlated to butyrate and TVFA (*p* < 0.05) and the relative abundance of *Lactobacillus* were positively correlated to acetate and isobutyrate (*p* < 0.05). The molar proportions of propionate and TVFA concentrations were negatively correlated to the relative abundance of *Rikenellaceae_RC9_gut_group* and *Ruminococcaceae_UCG002* (*p* < 0.05). Furthermore, isobutyrate concentrations were positively correlated to the relative abundance of *Lachnospiraceae_NK3A20_group*, *Erysipelotrichaceae_UCG006*, *f__Prevotellaceae_Unclassified*, *Erysipelotrichaceae_UCG009* and *Shuttleworthia* (*p* < 0.05). The molar proportions of isovalerate was positively related (*p* < 0.05) with *Lachnospiraceae_NK3A20_group*, but negatively correlated to *f__Erysipelotrichaceae_Unclassified* (*p* < 0.05). On the contrary, lactate concentration was positively correlated to the relative abundance of *f__Erysipelotrichaceae_Unclassified* (*p* < 0.05), whereas negatively correlated to the relative abundance of *Lachnospiraceae_NK3A20_group* and *Shuttleworthia* (*p* < 0.05). The ruminal pH was positively related (*p* < 0.05) with *f__Erysipelotrichaceae_Unclassified* and *Erysipelotrichaceae_UCG009*, but negatively correlated to the relative abundance of *Howardella* (*p* < 0.05). The molar proportion of acetate and propionate concentration was negatively correlated with the relative abundance of *Rikenellaceae_RC9_gut_group* and *Ruminococcaceae_UCG002* (*p* < 0.05). The concentrations of LPS was positively related (*p* < 0.05) with *Erysipelotrichaceae_UCG006*, *Acetitomaculum*, and *Erysipelotrichaceae_UCG009*, but negatively correlated to *Rikenellaceae_RC9_gut_group*, *Selenomonas_1*, and *Prevotellaceae_UCG003*. No significant correlations were found (*p* > 0.05) between the relative abundance of genera and valerate. 

## 4. Discussion

SARA is a consequence of feeding high grain diets to ruminant animals and it is characterized by a ruminal pH of <5.8 for more than 3 h daily [27]. In th present study, ruminal pH fell below 5.8 in control diet and this lasted more than 3 h (from 3 h to 6 h sampling times), indicating that SARA was induced successfully in sheep receiving the control diet. In addition, the concentration of free rumen LPS was 138,992.37 EU/mL for the control group in our experiment. Similar to our results, Pan et al. reported that the ruminal LPS concentration varied from 11,815 EU/mL in the 40% concentrate diet to 134,380 EU / mL when 65% concentrate was included in the diets of dairy cows [28]. Furthermore, the low-pH rumen conditions may increase the permeability of the gut to LPS, which could increase the LPS content in the peripheral blood during SARA. Khafipour et al. reported that a grain-based SARA challenge increased plasma LPS from <0.05 (Control) to 0.52 EU/mL (SARA) [29]. Our results revealed that rumen LPS entered the blood through the rumen epithelial barrier and caused an increase in the concentration of LPS in the blood. According to our results, we successfully established SARA model and inflammatory stress model in our study.

Numerous studies have shown that BSS can decrease the levels of TNF-α, IL-6, and IL-1β in intestinal tissue of mice by inhibiting the binding of LPS to TLR4 in the NF-kB pathway [30]. In the current research, we investigated the effect of dietary BSS supplementation on proinflammatory cytokines in serum during SARA induced by high grain diet and found that dietary BSS at levels of 1 g/kg lowered the levels of TNF-α, IL-6, and IL-1β, but increased the levels of TLR4. Our results suggest that BSS could alleviate the chronic inflammatory response to high-grain feeding by reducing the secretion of inflammatory factors in ruminants. 

In this study, we found that dietary BSS at levels of 0.5 and 1 g/kg in high grain diet could modify rumen fermentation through increasing ruminal pH and acetate content, and reducing the accumulation of lactate and free rumen free LPS in the rumen. In order to understand the potential mechanism of BSS on rumen fermentation, high-throughput sequencing was used to investigate the response of bacterial community to BSS supplementation under high grain feeding. 

It has been previously demonstrated that high grain diet altered the community of the ruminal bacterial microbiota, decreasing the abundance of Bacteroidetes (gram-negative species) and increasing the concentration of LPS under low pH [29,31]. Our observations revealed that dietary supplementation with BSS could improve the abundance of Bacteroidetes, this may be attributed to the increased ruminal pH and the decreased ruminal LPS. Moreover, no significant shifts detected in the abundance of Firmicutes. Interestingly, in the present study, the MBS and HBS groups had lower abundance of Cyanobacteria compared with CON group. Cyanotoxins produced by cyanobacteria may pose a threat to animals due to their widespread occurrence in both fresh and sea waters [32]. In spite of a recent study indicate that rumen microorganisms may degrade several cyanotoxins [33], the threat from cyanotoxins cannot be ignored. Our results revealed that BSS might inhibit the growth of cyanobacteria, since its antibacterial activity has been documented [34,35].

Previous studies revealed that genus *Selenomonas* ferments lactate to acetate and propionate and high grain feeding decreases the abundance of *Selenomonas* due to low-pH inhibits the growth of *Selenomonas* [36,37]. In the present study, we found that HBS group had a higher proportion of genus *Selenomonas_1*. Thus, the increasing *Selenomonas* promoted the consumption of lactate and increasing ruminal pH since no significant effect was observed in the VFA concentrations by BSS supplementation. This study also reveals that BSS treatment increased the abundance of genera *Prevotella*, such as *Prevotella_1*, *f__Prevotellaceae_Unclassified*, *Prevotella_7*, *Prevotellaceae_UCG-003*, *Prevotellaceae_UCG004*. It has been reported that high grain feeding decreased the abundance of the taxa *Prevotella*, which may be related to the lower pH as *Prevotella* is known to be sensitive to environmental pH [38]. Thus, the enrichment of the genera *Prevotella* in our study might be explained from two aspects. One plausible explanation is that the increasing *Selenomonas* helped to increase ruminal pH, thus enhanced the activity of *Prevotella*. The other plausible explanation might be BSS is a potential growth promoter for *Prevotella*. However, little information about BSS effect on *Prevotella* was reported in literature, which indicates the need for more research on the interactions between *Prevotella* and BSS. Previous studies showed that most of the *Prevotella* strains ferment glucose to produce acetate and succinate [39]. In the current study, we observed that there was no change in the proportions of predominant acetate-producing bacteria, including *Ruminococcus**_2*, *Ruminococcaceae_NK4A214_group*. These reasons may help explain the higher acetate concentrations as well as the increased *Succiniclasticum* which was the primary succinate utilizing bacteria. Most genera of *Lachnospiraceae* have been associated with the production of butyrate [40,41]. In our study, we observed that the abundance of *Lachnospiraceae_NK3A20_group* in HBS group was higher than other group and positively correlated to butyrate. Above all, dietary BSS Supplementation inhibited the growth of *Lachnospiraceae_NK3A20_group*, thus decreased butyrate production. Interestingly, *Rikenellaceae_RC9_gut_group* accounted for 11.3% of total bacterial community in HBS group and increased significantly compared with CON group. Besides, *Rikenellaceae_RC9_gut_group* is classified in the *Rikenellaceae* family and members of these family are hydrogen-producing bacteria that selectively neutralize cytotoxic reactive oxygen species [42]. It has been reported that endogenous hydrogen mediates suppression of proinflammatory cytokines, especially, IL-1β, TNF-α, IL-6 in inflammatory tissues [43]. Liu et al. (2015) reported that the genera *Unclassified Rikenellaceae* was reduced in the high grain feeding goats compared with hay diet [44]. Our results suggest that BSS may have the potential to reduce oxidative stress under high grain diet in ruminants. In addition, this study also shows that BSS supplementation increased the abundance of *Erysipelotrichaceae*, *Shuttleworthia*, and *Syntrophococcus*. However, these genera have a lower abundance of total rumen bacteria as well as their metabolic and functional significance in the ruminal ecosystem is unknown, the reasons for the altered status of genera by BSS supplementation are unclear.

Nevertheless, it is important to note that BSS attenuates inflammatory stress and modifies rumen fermentation in a dose-dependent manner in the present study. We found that dietary BSS at levels of 0.5 and 1 g/kg reduced the accumulation of ruminal lactate and free rumen LPS and inhibited the release of inflammatory factors (IL-6, IL-1β) and TNF-α in the blood. However, dietary BSS at levels of 1 g/kg has a higher concentration of NH_3_-N, which might represent the lower efficiency of dietary N transformation to microbial N. Taken together, we suggest that the dietary supplement level of BSS is 0.5g/kg DM.

## 5. Conclusions 

In summary, the results from this study implied that dietary BSS supplementation alleviates the chronic inflammatory response induced by high-grain feeding in sheep, that is associated with decreasing levels of serum TNF-α, IL-6, and IL-1β. In addition, BSS supplementation to high grain diet improves rumen fermentation through increasing ruminal pH and acetate content, and reducing the accumulation of lactate and ruminal LPS in a dose-dependent manner. Dietary BSS supplementation increased the abundance of the phylum Bacteroidetes, which reduces the release of LPS. Dietary BSS supplementation increased the abundance of the genera Prevotella_1, which may contribute to the production of acetic acid in the rumen. Dietary BSS supplementation also increased the abundance of the genera Selenomonas_1, which may reduce the accumulation of lactate. After receiving BSS treatment, the abundance of Rikenellaceae_RC9_gut_group increased significantly, which indicated that BSS had the potential to alleviate oxidative stress. Dietary BSS decreased the abundance of Lachnospiraceae_NK3A20_group, which may reduce the production of butyric acid. In addition, the BSS dietary supplementation at the level of 0.5 g/kg is recommended in sheep. Overall, BSS has potential for use as feed additive to alleviate inflammatory response and to modify ruminal fermentation in a high grain intensive ruminant production.

## Figures and Tables

**Figure 1 animals-10-00171-f001:**
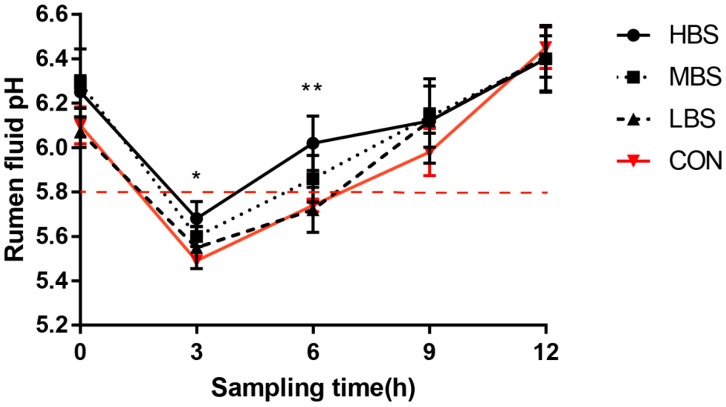
Mean ruminal pH values with different levels of β-sitosterol (BSS) supplementation. CON: basal diet without supplement; LBS: basal diet low supplemented with 0.25 g of BSS/kg of dry matter intake (DMI); MBS: basal diet medium supplemented with 0.5 g of BSS/kg of DMI; HBS: basal diet high supplemented with 1 g of BSS/kg of DMI. SEM, standard error of the mean. Data were expressed as mean ± SE. * indicates differences between dietary treatments at *p* < 0.05.; ** indicates differences between dietary treatments at *p* < 0.01.

**Figure 2 animals-10-00171-f002:**
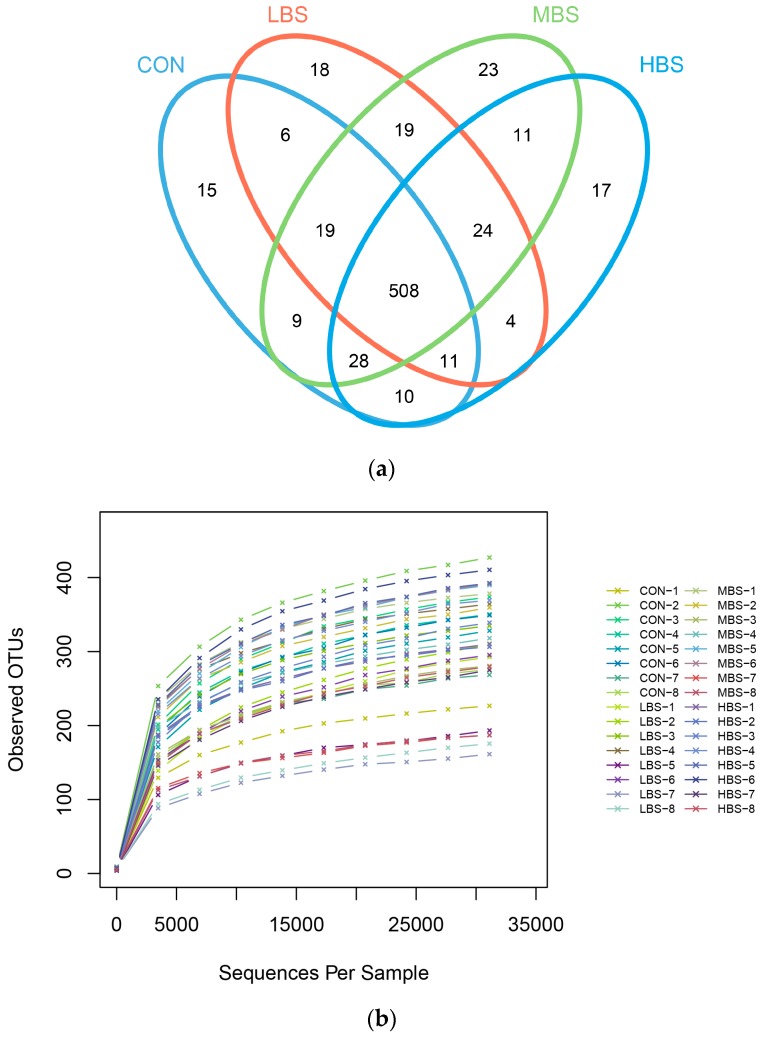
Ruminal microbial operational taxonomic units (OTUs) with different levels of BSS supplementation. CON: basal diet without supplement; LBS: basal diet low supplemented with 0.25 g of BSS/kg of DMI; MBS: basal diet medium supplemented with 0.5 g of BSS/kg of DMI; HBS: basal diet high supplemented with 1 g of BSS/kg of DMI (**a**) Venn diagram of ruminal bacterial OTUs. The number of unique OTUs were represented by the unoverlapped portion of Venn diagram for each group. (**b**) Bacterial rarefaction curves based on OTUs were used to assess the depth of coverage for each sample.

**Figure 3 animals-10-00171-f003:**
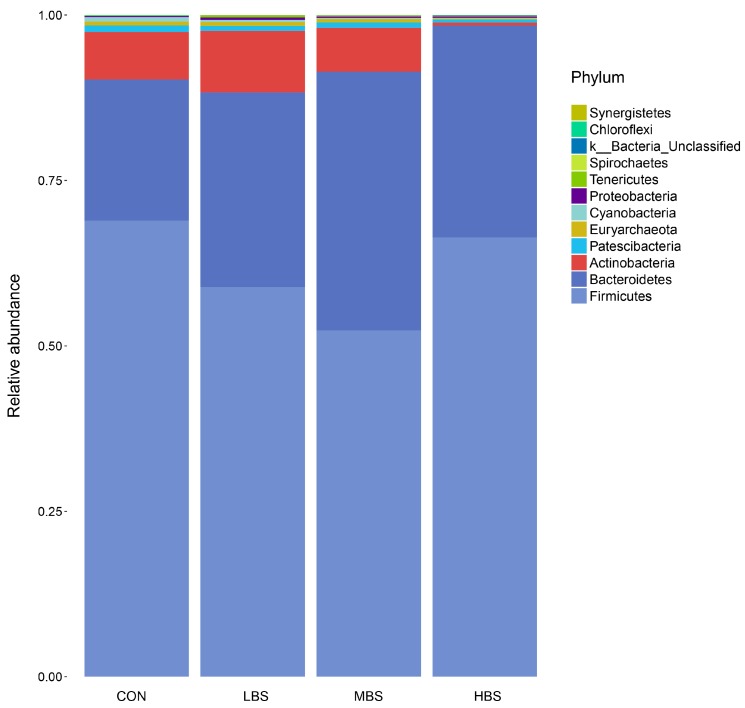
Percentage composition of the total phyla in rumen fluid. CON: basal diet without supplement; LBS: basal diet low supplemented with 0.25 g of BSS/kg of DMI; MBS: basal diet medium supplemented with 0.5 g of BSS/kg of DMI; HBS: basal diet high supplemented with 1 g of BSS/kg of DMI.

**Figure 4 animals-10-00171-f004:**
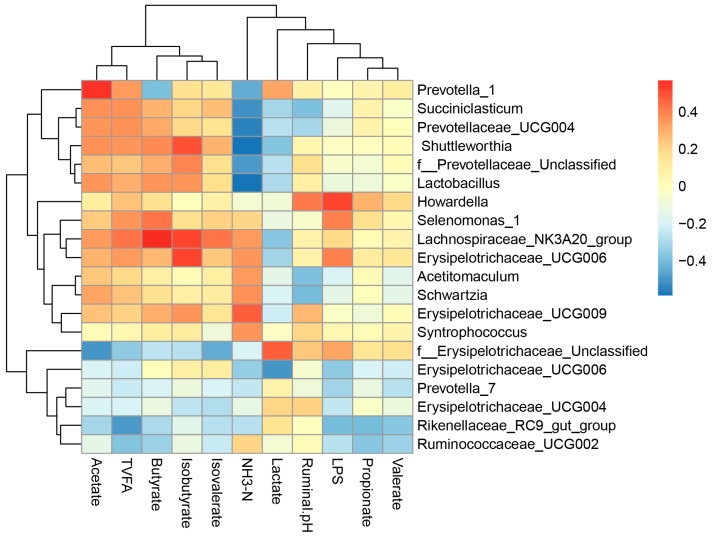
Coefficients of correlation between relative abundances of bacteria genera, and ruminal fermentation parameters.

**Table 1 animals-10-00171-t001:** Ingredients, composition, and nutrient contents of the experimental diet (DM basis) (%).

Item (% of DM ^1^, Unless Otherwise Stated)	Compositions of Experimental Diet
Ingredient	
Oat grass hay	11.7
Alfalfa hay	6.9
Grounded corn	62.7
Soybean meal	11.9
CaHPO _4_	1.1
NaCl	0.4
Premix ^2^	5.3
Total	100.0
Nutrient levels	
ME ^3^ (MJ/kg of DM ^1^)	11.6
CP ^4^	10.0
NDF ^5^	32.5
NFC ^3^	66.0
NFC/NDF	2.0
Ca	0.6
P	0.3

^1^ DM: dry matter. ^2^ Premix contained (per kg): 1000 mg of choline, 190 mg of nicotinic acid, 10.5 mg of I, 6 mg of Se, 800 mg of Mn, 2000 mg of Zn, 550 mg of Cu, 6.5 mg of Co, 3000 mg of Fe, 130 IU of Vitamin E, 150,000 IU of Vitamin D_3_, 70,000 IU of Vitamin A. ^3^ ME (metabolizable energy) and NFC (non-fiber carbohydrate) were calculated in reference to NRC (2001), while the other nutrition levels were measured values. ^4^ CP: Crude protein ^5^ NDF: Neutral detergent fiber.

**Table 2 animals-10-00171-t002:** Effect of dietary β-sitosterol (BBS) supplementation on cytokines, TLR4, and LPS in the blood of sheep.

Items	Treatments ^2^				SEM ^3^	*p*-Value
	CON	LBS	MBS	HBS		
TNF-α (ng/L) ^1^	347.48 ^a^	342.11 ^ab^	323.35 ^b^	324.60 ^b^	6.86	0.044
IL-6 (ng/L) ^4^	29.38 ^a^	29.43 ^a^	26.86 ^b^	26.82 ^b^	0.62	0.005
IL-8 (ng/L) ^5^	254.17	252.96	251.49	245.19	7.19	0.611
IL-1β (ng/L) ^6^	119.21 ^a^	106.69 ^ab^	104.70 ^b^	105.87 ^b^	2.14	<0.001
TLR4 (ng/L) ^7^	34.94 ^b^	36.38 ^a^	33.32 ^b^	35.78 ^b^	0.50	0.002
LPS (EU/mL)	0.20 ^a^	0.15 ^ab^	0.06^c^	0.06 ^c^	0.02	<0.001

^1^ TNF-α = tumor necrosis factor α. ^2^ CON: basal diet without supplement; LBS: basal diet low supplemented with 0.25 g of BSS/kg of DMI; MBS: basal diet medium supplemented with 0.5 g of BSS/kg of DMI; HBS: basal diet high supplemented with 1 g of BSS/kg of DMI. ^3^ SEM, standard error of the mean. ^a,b,c^ Means within a row with different superscripts are different at *p* < 0.05. ^4^ IL-6 = interleukin-6. ^5^ IL-8 = interleukin-8. ^6^ IL-1β = interleukin-1β. ^7^ TLR4 = Toll-like receptor 4.

**Table 3 animals-10-00171-t003:** Effect of dietary β-sitosterol (BBS) supplementation on rumen fermentation parameters of sheep (values averaged across sampling times).

Items	Treatment ^3^				SEM ^4^	*p*-Value
	CON	LBS	MBS	HBS		
Ruminal pH	5.75 ^b^	5.72 ^b^	5.86 ^ab^	6.02 ^a^	0.08	0.006
NH_3_-N (mg/dL)	5.73 ^b^	7.39 ^ab^	9.63 ^a^	10.19 ^a^	1.03	0.021
Lactate (mmol/L)	1.12 ^a^	0.96 ^ab^	0.65 ^b^	0.67 ^b^	0.12	0.023
LPS (×10^3^ EU/mL)	138.99 ^a^	92.45 ^ab^	66.41 ^b^	56.07 ^b^	4.80	<0.001
TVFA (mmol/L) ^1^	109.32	114.89	109.79	114.34	0.81	0.074
Acetate (%) ^2^	48.29 ^b^	49.67 ^b^	58.27 ^a^	56.60 ^a^	1.84	<0.001
Propionate (%) ^2^	28.34 ^a^	29.72 ^a^	24.66 ^ab^	21.09 ^b^	1.89	0.017
Isobutyrate (%) ^2^	0.84	0.66	0.77	1.25	0.16	0.060
Butyrate (%) ^2^	19.30 ^a^	17.28 ^ab^	14.13 ^b^	15.92 ^ab^	1.20	0.037
Isovalerate (%) ^2^	1.37 ^ab^	1.07 ^b^	1.03 ^b^	1.61 ^a^	0.15	0.037
Valerate (%) ^2^	1.85 ^a^	1.59 ^a^	1.14 ^b^	1.53 ^a^	0.12	0.004

^1^ TVFA: total volatile fatty acids. ^2^ Acetate (%), propionate (%), isobutyrate (%), butyrate (%), isovalerate (%), and valerate (%) imply the molar proportion of each to that of the TVFA. ^3^ CON: basal diet without supplement; LBS: basal diet low supplemented with 0.25 g of BSS/kg of DMI; MBS: basal diet medium supplemented with 0.5 g of BSS/kg of DMI; HBS: basal diet high supplemented with 1 g of BSS/kg of DMI. ^4^ SEM, standard error of the mean. ^a,b^ Means within a row with different superscripts are different at *p* < 0.05.

**Table 4 animals-10-00171-t004:** Number of observed species, richness and diversity indices in ruminal samples from each dietary treatment.

Items	Treatments ^1^				SEM ^2^	*p*-Value
	CON	LBS	MBS	HBS		
OUT ^3^	331.13 ^a^	272.38 ^b^	317.25 ^ab^	342.25 ^a^	17.15	0.043
ACE index ^4^	377.38 ^ab^	329.63 ^b^	361.55 ^ab^	392.92 ^a^	13.74	0.023
Chao1 index	384.70 ^ab^	333.00 ^c^	359.23 ^b^	405.78 ^a^	14.07	0.008
Shannon index	4.65	4.17	5.13	4.35	0.30	0.149
Simpson index	0.85	0.82	0.90	0.78	0.06	0.292
Coverage (%)	99.8	99.8	99.8	99.8	0.00	0.895

^1^ CON: basal diets without supplement; LBS: basal diets low supplemented with 0.25 g of BSS/kg of DMI; MBS: basal diet medium supplemented with 0.5 g of BSS/kg of DMI; HBS: basal diet high supplemented with 1 g of BSS/kg of DMI. ^2^ SEM, standard error of the mean. ^3^ OTU, operational taxonomic units. ^4^ ACE, abundance-based coverage estimator ^a,b,c^ Means within a row with different superscripts are different at *p* < 0.05.

**Table 5 animals-10-00171-t005:** Effects of β-sitosterol (BBS) supplementation on relative abundance in rumen fluid at phylum level.

Phylum	Treatment ^1^				SEM ^2^	*p*-Value
	CON	LBS	MBS	HBS		
Firmicutes	68.94	58.88	52.32	66.38	5.81	0.200
Bacteroidetes	21.34 ^b^	29.43 ^ab^	39.11 ^a^	32.04 ^ab^	4.38	0.041
Actinobacteria	7.17	9.30	6.62	0.46	4.43	0.543
Patescibacteria	0.99	0.76	0.83	0.42	0.22	0.324
Euryarchaeota	0.58	0.58	0.51	0.13	0.26	0.292
Cyanobacteria	0.66 ^a^	0.33 ^ab^	0.16 ^b^	0.13 ^b^	0.22	0.046
Proteobacteria	0.18	0.33	0.23	0.24	0.09	0.675
Tenericutes	0.02	0.31	0.07	0.07	0.15	0.435
Spirochaetes	0.05	0.04	0.14	0.09	0.028	0.097
k__Bacteria_Unclassified	0.02	0.03	0.02	0.06	0.02	0.124
Chloroflexi	0.06	0.01	0.03	0.02	0.02	0.361
Synergistetes	0.005	0.011	0.006	0.008	0.03	0.465

^1^ CON: basal diets without supplement; LBS: basal diets low supplemented with 0.25 g of BSS/kg of DMI; MBS: basal diet medium supplemented with 0.5 g of BSS/kg of DMI; HBS: basal diet high supplemented with 1 g of BSS/kg of DMI. ^2^ SEM, standard error of the mean. ^a, b^ Means within a row with different superscripts are different at *p* < 0.05.

**Table 6 animals-10-00171-t006:** Effects of β-sitosterol (BBS) supplementation on relative abundances of bacterial genera in rumen fluid using 16S rRNA sequencing ^1^ (%).

Phylum	Family	Genus	Treatments ^2^				SEM ^3^	*p*-Value
			CON	LBS	MBS	HBS		
Bacteroidetes	Prevotellaceae	*Prevotella_1*	7.06 ^b^	6.53 ^b^	9.18 ^a^	13.84 ^a^	2.09	0.041
		*f__Prevotellaceae_Unclassified*	0.41 ^b^	8.33 ^a^	2.62 ^ab^	0.26 ^b^	1.99	0.030
		*Prevotella_7*	0.04 ^b^	0.37 ^b^	1.77 ^a^	0.23 ^b^	0.41	0.028
		*Prevotellaceae_UCG-003*	0.23 ^b^	0.19 ^b^	0.51 ^ab^	0.80 ^a^	0.17	0.045
		*Prevotellaceae_UCG004*	0.07 ^b^	0.26 ^ab^	0.52 ^a^	0.06 ^b^	0.12	0.018
	Rikenellaceae	*Rikenellaceae_RC9_gut_group*	2.67 ^b^	3.45 ^b^	7.21 ^ab^	11.33 ^a^	1.75	0.008
Firmicutes	Erysipelotrichaceae	*Erysipelotrichaceae_UCG006*	4.85 ^a^	3.23 ^ab^	0.46 ^b^	0.004 ^b^	1.16	0.022
		*Erysipelotrichaceae_UCG-004*	0.25 ^b^	0.21 ^b^	0.77 ^a^	0.33 ^b^	0.17	0.049
		*Erysipelotrichaceae_UCG-009*	0.07 ^b^	0.99 ^a^	0.20 ^b^	0.05 ^b^	0.19	0.005
		*f__Erysipelotrichaceae_Unclassified*	0.19 ^a^	0.19 ^a^	0.09 ^b^	0.08 ^b^	0.02	0.017
	Veillonellaceae	*Selenomonas_1*	0.32 ^b^	0.27 ^b^	1.04 ^b^	3.56 ^a^	0.74	0.015
		*Succiniclasticum*	0.23 ^b^	0.25 ^b^	1.75 ^a^	0.42 ^b^	0.32	0.009
	Unclassified	*Howardella*	0.60 ^a^	0.28 ^ab^	0.18 ^b^	0.13 ^b^	0.13	0.038
	Lachnospiraceae	*Lachnospiraceae_NK3A20_group*	5.09 ^ab^	8.03 ^a^	4.19 ^ab^	1.77 ^b^	1.60	0.046
		*Acetitomaculum*	1.42 ^a^	0.63 ^ab^	1.09 ^ab^	0.44 ^b^	0.27	0.043
		*Shuttleworthia*	0.16 ^b^	0.52 ^a^	0.32 ^ab^	0.09 ^b^	0.11	0.044
		*Syntrophococcus*	0.13 ^b^	0.31 ^a^	0.14 ^b^	0.06 ^b^	0.04	0.005
	lactobacillaceae	*Lactobacillus*	0.11 ^b^	0.71 ^a^	0.35 ^ab^	0.03 ^b^	0.15	0.010
	Selenomonadaceae	*Schwartzia*	0.02 ^b^	0.07 ^b^	0.49 ^a^	0.02 ^b^	0.10	0.010
	Ruminococcaceae	*Ruminococcaceae_UCG002*	0.02 ^b^	0.03 ^b^	0.14 ^ab^	0.38 ^a^	0.09	0.032

^1^ Only bacterial genera (accounted for ≥0.1% in at least one of the samples) that affected by treatments were listed. ^2^ CON: basal diets without supplement; LBS: basal diets low supplemented with 0.25 g of BSS/kg of DMI; MBS: basal diet medium supplemented with 0.5 g of BSS/kg of DMI; HBS: basal diet high supplemented with 1 g of BSS/kg of DMI. ^3^ SEM, standard error of the mean. ^a,b^ Means within a row with different superscripts are different at *p* < 0.05.

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
