# Peer review of "β-Sitosterol Attenuates High Grain Diet-Induced Inflammatory Stress and Modifies Rumen Fermentation and Microbiota in Sheep"

_animals, 2020, doi:10.3390/ani10010171_

Round 1
Reviewer 1 Report
L54-56 The sentence need be rewrote.
L87 the specific feeding time needed.
L90 the sentence need be rewrote.
L91 delete "chemical"
L92 ME in table 1 is a calculated value too, should be noted below the table. besides, the content of co (650mg) in the premix is wrong, need be checked again.
L156 which data were subjected to ANOVA need to be clear
L186 ruminal pH in table 3 should not be significant because the p value is over 0.05.
L329 the name of "HG" is needed
L358-359 it is inexact to say fermentation is inhibited by BSS because the total VFA and diversity of microbiota are not decreased.
L362 it is also inaccurate to say. I suggest the author give the OTUs at table 4. it can be a evidence for the number of bacteria.
Author Response
Point 1: L54-56 The sentence need be rewrote.
Response 1: Thanks for the reviewer’s kind suggestion. The revised sentence is as follows: Several diet strategies, such as sodium bicarbonate buffer [6], dicarboxylic acids [7], monensin iono-phones [8] and probiotics [9], have been used to prevent the occurrence of SARA. The details can be found in Line 132-133
Point 2: L87 the specific feeding time needed.
Response 2: Thanks for the reviewer’s kind suggestion. The sheep were feed twice daily at 6:00 and 18:00 and one-half of the allowed daily ration at each feeding. The details can be found in Line 164-166
Point 3: L90 the sentence need be rewrote.
Response 3: Thanks for the reviewer’s kind suggestion. We have rewrote the sentence and the details can be found in Line 167-216
Point 4: L91 delete "chemical".
Response 4: Thanks for the reviewer’s kind suggestion. According to your advices, we have deleted "chemical" in revised manuscript.
Point 5: L92 ME in table 1 is a calculated value too, should be noted below the table. besides, the content of co (650mg) in the premix is wrong, need be checked again.
Response 5: Thanks for the reviewer’s kind suggestion. We have added a note about “ME” below the table. The details can be found in Line 223. In addition, we are sorry for this mistake. The content of co in the premix should be 6.5mg, which is a mistake in writing. The details can be found in Line 222.
Point 6: L156 which data were subjected to ANOVA need to be clear
Response 6: Thanks for the reviewer’s kind suggestion. Cytokines concentrations, rumen fermentation parameters, diversity index, and bacterial abundance were subjected to an ANOVA analysis. The details can be found in Line 439-440.
Point 7: L186 ruminal pH in table 3 should not be significant because the p value is over 0.05.
Response 7: Thanks for the reviewer’s kind suggestion. P value of ruminal pH should be 0.0061 instead of 0.061, which is a mistake in our writing. The revised details can be found in Line 488 (Table 3).
Point 8: L329 the name of "HG" is needed
Response 8: Thanks for the reviewer’s kind suggestion. We have replaced “HG” with “high grain”. The revised details can be found in Line 808.
Point 9: L358-359 it is inexact to say fermentation is inhibited by BSS because the total VFA and diversity of microbiota are not decreased.
Response 9: Thanks for the reviewer’s kind suggestion. We realized that it was not accurate to say that BSS inhibited rumen fermentation, and we rewrote this paragraph. The revised details can be found in Line 982-988.
Point 10: L362 it is also inaccurate to say. I suggest the author give the OTUs at table 4. it can be a evidence for the number of bacteria.
Response 10: Thanks for the reviewer’s kind suggestion. We have supplemented the data of OTUs at Table 4. The revised details can be found in Line 653 (Table 4).

Reviewer 2 Report
I have reviewed the manuscript entitled "β‐Sitosterol attenuates high grain diet-induced inflammatory stress and modifies rumen fermentation by regulating ruminal microbiota in sheep" which assess the potential of a particular plant compound to modify ruminal parameter due to acidosis in high grain diets. The study is interesting and with potential high impact to reduce the use of antibiotics to treat health problems in ruminants.
General comment
Apart from some moderate English style defects, the study is well written, and the methodology is clearly presented. However, there are some flaws in the Results section that have apparently misled the discussion and conclusions. There are some mistakes on the format of tables, figures and subtitles that should be checked.
Specific comments
Please make sure you introduce each acronym in its first use and you keep the consistency in their use throughout the manuscript, e.g. NCF and NDF in L28, DMI in L82.
Abstract
L39: this recommendation is not stated in the Discussion nor in the Conclusions section. This is supposed to be the summary of the manuscript, therefore, all the statements done here should be present in the main text.
Keywords
It would improve the searches appearance of your work if you use keywords different than those included in the title. I would recommend replacing the repeated expression by new ones.
Materials and Methods
L88, L89: I would recommend replacing “orts” by “feed refusal” since the former is not a common word in English
Why is the supplement not fed along with the feed instead of via rumen canula?
How much feed is fed to the sheep? The authors state a level of refusal but do not provide values or ranges for the feed offered and consumed.
Table 1 – Is the ME expressed per kg DM or per kg of fresh feed?
Results
The authors have not stated whether the SARA or the inflammatory stress did occur under the CON diet. Therefore, it is difficult to infer about the effect of the BSS supplementation on the attenuation of this syndrome. Please explicitly state whether or not the intended effect of the high-grain diet (SARA or inflammatory stress) was actually achieved.
This section has a major error of interpretation that misled all the other results description and the discussion and conclusions: the authors stated that “As shown in Table 3, BSS supplementation increased (P < 0.05) ruminal pH value”, however the p value for pH was greater than 0.05 (it was 0.061). Therefore, given that “Significance was declared at P < 0.05”, the numerical differences in pH did not reach statistical significance. Therefore, all the results and discussion should be amended to account for the lack of effect of the BSS supplementation on the ruminal pH.
The rest of the results are not properly described since they are very much simplified. For instance, the authors state that “BSS supplementation increased the molar proportions of acetate, whereas the concentrations of lactate, and the molar proportions of propionate, butyrate and valerate were decreased significantly compared with the CON treatment”; however, according to the superscript letters in Table 3, the concentration of propionate only decreased with MBS and HBS (it is not correct saying that BSS supplementation decreased this VFAs since the LBS did not differ from the control), the concentration of butyrate only differed from the from the control in MBS, and the concentrations of valerate was only different from the CON under MBS dose. Moreover, assuming the 1.12 value (lactate concentration under CON) is accompanied by a letter “a” (not b), this concentration only differed from those under MBS and HBS. Also, it is not correct saying that the BSS supplementation decreased NH3 since it seems it was increased. There is not statement about the effect of BSS on isovalerate.
L197 “13” should be replaced by “23” according to Figure 1a
L216-218 Please describe in more detail the results in Table 4.
Table 4: Should the value 333.0 (Chao1inex under LBS) be accompanied by a “b”? Please check.
L238: what do the authors refer with “the study period”? There was not initial sampling (baseline) therefore you cannot refer to a change across time. Please rephrase accordingly.
L241: LBS should be replaced by MBS
L237-245 Please check carefully the accuracy of the description of results (as the example mentioned just above)
L246 The title of the Table 6 (it is not 5) should be on above the table.
L248-253: Is this the footnote? Please check format as it looks as the main text.
Discussion
The first nine lines of the discussion are almost a repetition of the introduction. Please remove repeated information. Also please include at the beginning of the discussion whether the intended syndrome (SARA or inflammatory stress) was achieved and provide evidence of that (threshold for the indicators).
All this section must be re-written since it is based in the fact that the ruminal pH was increased with the BSS supplementation but that was not the case.
L329: what is “HG”?
L353: can the authors elaborate about the consequences of an accumulation of hydrogen in the rumen?
Conclusions
This section should reflect the findings. Did the stress occur? This should be also amended to reflect the fact that the pH was not affected by BSS.
Author Response
Point 1: Please make sure you introduce each acronym in its first use and you keep the consistency in their use throughout the manuscript, e.g.
Response 1: Thanks for the reviewer’s kind suggestion. We have revised these mistakes and the revised details can be found in Line 29/159.
Point 2: Abstract L39: this recommendation is not stated in the Discussion nor in the Conclusions section. This is supposed to be the summary of the manuscript, therefore, all the statements done here should be present in the main text. L39:
Response 2: Thanks for the reviewer’s kind suggestion. We have stated in the discussion the revised details can be found in Line 982-988.
Point 3: It would improve the searches appearance of your work if you use keywords different than those included in the title. I would recommend replacing the repeated expression by new ones.
Response 3: Thanks for the reviewer’s kind suggestion. We have modified the keywords as follows: β-sitosterol; subacute ruminal acidosis; inflammatory response; bacterial community; pyrosequencing; sheep. The revised details can be found in Line 42-43.
Point 4: L88, L89: I would recommend replacing “orts” by “feed refusal” since the former is not a common word in English
Response 4: Thanks for the reviewer’s kind suggestion. We have replaced “orts” by “feed refusal”. The details can be found in Line 89.
Point 5: Why is the supplement not fed along with the feed instead of via rumen canula?
Response 5: Thank you for your questions. In our experiment, we established four levels of dietary supplement BBS. Due to the small amount of low dose, in order to avoid different levels of BSS intake in different treatment groups, we supplement BSS via rumen canula instead of direct feeding, which can reduce the error of the experiment.
Point 6: How much feed is fed to the sheep? The authors state a level of refusal but do not provide values or ranges for the feed offered and consumed.
Response 6: Thanks for the reviewer’s kind questions and suggestions. The DMI is 1.6 kg dry matter (DM) per sheep per day and we offered the diet (1.8 kg dry matter (DM) per sheep per day) was offered in equal amounts at 08:00 and 18:00 daily and 5 to 10% feed refusal is allowed. We have revised them and the details can be found in Line 88-91.
Point 7: Table 1 – Is the ME expressed per kg DM or per kg of fresh feed?
Response 7: Thanks for the reviewer’s kind suggestion. The ME is expressed per kg of DM. The revised details can be found in Line 96 (Table 1).
Point 8: The authors have not stated whether the SARA or the inflammatory stress did occur under the CON diet. Therefore, it is difficult to infer about the effect of the BSS supplementation on the attenuation of this syndrome. Please explicitly state whether or not the intended effect of the high-grain diet (SARA or inflammatory stress) was actually achieved.
Response 8: Thanks for the reviewer’s kind suggestion. We have added the data of ruminal and plasma LPS concentration in the table 2 and 3, and added the curve of rumen fluid pH in four groups of sheep from 0 to 12h after morning feeding. We have discussed at the beginning of the discussion whether the SARA model successfully induced. The revised details can be found in Line 339-352.
Point 9: This section has a major error of interpretation that misled all the other results description and the discussion and conclusions: the authors stated that “As shown in Table 3, BSS supplementation increased (P < 0.05) ruminal pH value”, however the p value for pH was greater than 0.05 (it was 0.061). Therefore, given that “Significance was declared at P < 0.05”, the numerical differences in pH did not reach statistical significance. Therefore, all the results and discussion should be amended to account for the lack of effect of the BSS supplementation on the ruminal pH.
Response 9: Thanks for the reviewer’s kind suggestion. We are sorry for this mistake. P value of ruminal pH should be 0.0061 instead of 0.061, which is a mistake in our writing. The revised details can be found in Line 221 (Table 3).
Point 10: The rest of the results are not properly described since they are very much simplified. For instance, the authors state that “BSS supplementation increased the molar proportions of acetate, whereas the concentrations of lactate, and the molar proportions of propionate, butyrate and valerate were decreased significantly compared with the CON treatment”; however, according to the superscript letters in Table 3, the concentration of propionate only decreased with MBS and HBS (it is not correct saying that BSS supplementation decreased this VFAs since the LBS did not differ from the control), the concentration of butyrate only differed from the from the control in MBS, and the concentrations of valerate was only different from the CON under MBS dose. Moreover, assuming the 1.12 value (lactate concentration under CON) is accompanied by a letter “a” (not b), this concentration only differed from those under MBS and HBS. Also, it is not correct saying that the BSS supplementation decreased NH3 since it seems it was increased. There is not statement about the effect of BSS on isovalerate.
Response10: Thanks for the reviewer’s kind suggestion. We have supplemented the description of the changes of rumen fermentation parameters and the revised details can be found in Line 213-220.
Point 11: L197 “13” should be replaced by “23” according to Figure 1a
Response 11: Thanks for the reviewer’s kind suggestion. We are sorry for this mistake. We have replaced “13” by “23”. The revised details can be found in Line 586.
Point 12: L216-218 Please describe in more detail the results in Table 4.
Response 12: Thanks for the reviewer’s kind suggestion. We have supplemented the description of the results in Table 4. The revised details can be found in Line 649-651.
Point 13: Table 4: Should the value 333.0 (Chao1 inex under LBS) be accompanied by a “b”? Please check.
Response 13: Thanks for the reviewer’s kind suggestion. We are sorry for this mistake. We have corrected this error in the revised manuscript. The details can be found in Line 652 (Table 4).
Point 14: L238: what do the authors refer with “the study period”? There was not initial sampling (baseline) therefore you cannot refer to a change across time. Please rephrase accordingly.
Response 14: Thanks for the reviewer’s kind suggestion. The revised statement is as follows: Of the 53 genus observed with over 0.1% of relative abundance, 20 changed significantly between the control and BSS treatments (Table 6). The details can be found in Line 710-711.
Point 15: L241: LBS should be replaced by MBS
Response 15: Thanks for the reviewer’s kind suggestion. We are sorry for this mistake. We have replaced “LBS” by “MBS”. The details can be found in Line 715.
Point 16: L237-245 Please check carefully the accuracy of the description of results (as the example mentioned just above) L237-245
Response 16: Thanks for the reviewer’s kind suggestion. We checked the accuracy of the description of results again and found some errors, so we supplemented and improved the interpretation of the results in the revised manuscript and the details can be found in Line 710-720.
Point 17: L246 The title of the Table 6 (it is not 5) should be on above the table.
Response: Thanks for the reviewer’s kind suggestion. We corrected this error in the revised manuscript and the details can be found in Line 721.
Point 17: L248-253: Is this the footnote? Please check format as it looks as the main text.
Response: Thanks for the reviewer’s kind suggestion. Lines 248 to 253 are footnotes to Table 6. We have modified the format in the revised manuscript and the details can be found in Line 723-727.
Point 18: Discussion: The first nine lines of the discussion are almost a repetition of the introduction. Please remove repeated information. Also please include at the beginning of the discussion whether the intended syndrome (SARA or inflammatory stress) was achieved and provide evidence of that (threshold for the indicators).
Response 18: Thanks for the reviewer’s kind suggestion. We have added the data of ruminal and plasma LPS concentration in the table 2 and 3, and added the curve of rumen fluid pH in four groups of sheep from 0 to 12h after morning feeding. We have discussed at the beginning of the discussion whether the SARA model successfully induced. The revised details can be found in Line 339-352.
Point 19: All this section must be re-written since it is based in the fact that the ruminal pH was increased with the BSS supplementation but that was not the case.。
Response 19: Thanks for the reviewer’s kind suggestion. P value of ruminal pH should be 0.0061 instead of 0.061, which is a mistake in our writing. The revised details can be found in Line 488 (Table 3).
Point 20: L329: what is “HG”?
Response 20: Thanks for the reviewer’s kind suggestion. We replaced “HG” by “high grain”. The revised details can be found in Line 825.
Point 21: L353: can the authors elaborate about the consequences of an accumulation of hydrogen in the rumen?
Response 21: Thanks for the reviewer’s question. Molecular hydrogen (H2) possesses the ability to selectively neutralize ONOO- and -OH, the most cytotoxic ROS, which can damage cellular macromolecules aggressively and indiscriminately. In inflammation process, H2 mediates suppression of proinflammatory cytokines, especially CCL2, IL-1β, TNF-α, IL-6 in inflammatory tissues. We speculated that BSS may BSS may have the potential to reduce oxidative stress under high grain diet in ruminants. (reference: Lactulose: An indirect antioxidant ameliorating inflammatory bowel disease by increasing hydrogen production. doi: 10.1016/j.mehy.2010.09.026)
Meanwhile, we revised them in the manuscript and the detail can be found in the Line 989-991.
Point 22: Conclusions:This section should reflect the findings. Did the stress occur? This should be also amended to reflect the fact that the pH was not affected by BSS.
Response 22: Thanks for the reviewer’s kind suggestion. we have revised them the details can be found in the Line 1007-1021.
